



# Bright and Blind Spots of Water Research in Latin America and the Caribbean

Alyssa J. DeVincentis[1], Hervé Guillon[1], Romina Díaz Gómez[1], Noelle K. Patterson[1], Francine van den Brandeler[2,3], Arthur Koehl[1], J. Pablo Ortiz-Partida[1,4], Laura E. Garza-Díaz[1], Jennifer Gamez-Rodríguez[1], Erfan Goharian[1,5], and Samuel Sandoval Solis[1]

[1]Department of Land, Air and Water Resources, University of California, Davis, USA
[2]University of Amsterdam, Amsterdam, Netherlands
[3]Greenleaf Communities, Chicago, USA
[4]Union of Concerned Scientists, Oakland, USA
[5]University of South Carolina, Columbia, USA

**Correspondence:** ajdevincentis@ucdavis.edu





**Abstract.** Water resources management is threatened by climatic, economic, and political pressures, and these challenges are on particular display in Latin America and the Caribbean. To assess the region's ability to manage water resources, we conducted an unprecedented literature review of over 20,000 multilingual research articles using machine learning and an understanding of the socio-hydrologic landscape. Results reveal that the region's vulnerability to water-related stresses, and drivers such as climate change, is compounded by research blind spots in niche topics (reservoirs and risk assessment) and sub-regions (Caribbean nations), and by its reliance on an individual country (Brazil). A regional bright spot, Brazil produces well-rounded water-related research but its regional dominance suggests that funding cuts there would impede scientifically-informed water management in the entire region.

## 1 Introduction

Despite being the world's most water-rich region, Latin America and the Caribbean (LAC) faces extreme weather events and a range of water-related stresses that are expected to worsen with climate change (UN-OCHA, 2020). It is critical that responses are based on sound science and contextualized. The Science of Science can help diagnose the state of water resources research in Latin America, and more specifically, its bright spots and blind spots.

Freshwater resources face mounting pressures, brought about by human population growth and urbanization (Jenerette and Larsen, 2006; Immerzeel et al., 2020), climate change (Gosling and Arnell, 2016), economic growth and consumption patterns (Mcdonald et al., 2014; O'dorico et al., 2018), and the spread of misinformation and mistrust in science (IPCC, 2014). LAC epitomizes these water challenges with its abundant yet unequally distributed water resources (DESA, 2019), mounting pollution, and the highest income inequality in the world (Varis et al., 2019). Marked disparities exist in terms of water availability (both quality and quantity), climate change vulnerability, degree and form of urbanization, conservation habits, and scientific productivity, which each affect water resources management (Ciocca and Delgado, 2017; Lyon et al., 2019). Countries with abundant water resources, Brazil for example, experience water scarcity due to a mismatch between water-rich areas and population centers (Formiga-Johnsson and Kemper, 2005), while, others, Argentina, Chile and Bolivia for example, face flooding and melting glaciers (Barros et al., 2015; Soruco et al., 2015; Masiokas et al., 2019), and yet others in Central America are increasingly devastated by hurricanes (Bárcena et al., 2020).

LAC is among the most urbanized regions in the world and these high-density areas face particular vulnerability to water quality and supply reliability (Kim and Grafakos, 2019). For example, São Paulo faced severe water shortages during a 2014 drought, while Mexico City has steadily and rapidly depleted its groundwater supply (Aguilar-Barajas et al., 2015). Urban pressures on water resources are compounded by inefficient farming practices, unregulated industries, and aging infrastructure across the region. Worldwide, and particularly in LAC, these water challenges are expected to intensify with climate change as variations in precipitation, temperature and evaporation threaten water availability for current and future water users (Dussaillant et al., 2019; Gesualdo et al., 2019; Zaninelli et al., 2019).

While uncertainty surrounds the reliability of water supplies to address water-related risks and meet future needs in LAC, water resources management is a relatively young field of study (Montanari et al., 2015) and suffers from a lack of interdisci-





plinary and integrative perspectives, common in the environmental sciences (Norgaard, 2008). Recent review papers are limited

to a geographic area (Owusu et al., 2016), individual components of the water budget, such as a watershed (Dobriyal et al., 2012), particular methodology (Plummer et al., 2012), specific water user (Ran et al., 2016), or small sample of documents (Endo et al., 2017).

Given these circumstances, it is critical to understand how the breadth of past water resources research across LAC contributes the scientific knowledge necessary for decision-making processes (Cvitanovic and Hobday, 2018). To assess the state

of water research in LAC, we used a Science of Science approach (Fortunato et al., 2018) and performed an unprecedented, multilingual review of available peer-reviewed literature to identify *bright spots* of scientific inquiry: topics and locations where water research is abundant, spread out, and well connected (Uzzi et al., 2013; Astudillo, 2016; Larivière et al., 2015). Conversely, *blind spots* of past research are defined as topics and locations where water resources are less-thoroughly studied. This approach identifies opportunities to enhance collaboration within the research community and ensure future research meets

societal needs.

To perform the literature review, we assembled a corpus of 20,000 water resources research articles in English, Spanish and Portuguese by querying online databases and modelled the topics of each article with Latent Dirichlet Allocation (Blei et al., 2003). Topics are groups of statistically co-occurring words, which we labeled based on four categories: 5 general research topics (e.g. physical sciences), 43 specific topics representing subfields of research (e.g. geochemistry), 17 water-budget topics

(e.g. reservoirs), and 13 method topics (e.g. remote sensing). For each article, topic probabilities are the probabilities that the article's content corresponds to each individual topic. In addition, we read a random subset of 2,000 papers from the corpus to validate results from the topic model and identify study location. We leveraged this manual reading process to identify the study location for the entire corpus using machine learning and used article metadata to generate a citation network of citing and cited references within the corpus.

To contextualize results from the literature review, we used publicly available data to statistically cluster countries into groups with similar social and hydrological systems, socio-economic metrics, and measures of water resources abundance and use. This clustering process allowed for more meaningful interpretation of subsequent results within and across countries. To further ground our results in the current research landscape, we invited 20,000 corresponding authors from our corpus to share their experiences through a survey focused on research discipline, accessibility, and connectivity. A total of 1,969 respondents

from 35 countries and a variety of disciplinary backgrounds completed this survey.

Bright spots and blind spots of water resources research in LAC were evaluated using three concepts: abundance, spread, and connectivity. Abundance was measured as research volume by country and by topic. Spread was estimated by topic normality across countries and articles, describing how close a topic's probability distribution is to the standard normal distribution. Connectivity was determined with a weighted citation network across countries and topics, describing the probability that a

specific node (country or topic) is cited by other nodes.

The rest of the article is organized as follows. The next section details how the data underlying this study were acquired. Section 3 presents the methods we used to associate metadata (e.g. topic, location) with each document in the corpus. In that section, we also introduce the specific metrics and methods used to infer bright and blind spots. Sections 4 exposes results





associated with metadata generation. Section 5 presents and discusses bright spots and blind spots. The last section summarizes
our findings.

## 2    Materials

In this section, we detail the specific process for corpus collection, retrieving socio-hydrologic data and survey.

### 2.1    Corpus collection

Our methodology for corpus collection consisted of four steps: (i) querying online databases; (ii) retrieving documents: (iii)
iteratively assessing quality of the corpus and correcting bias; and (iv) cleaning the corpus.

First, we defined the query to obtain water resources research about Latin American and Caribbean (LAC) countries. We
selected a peer-reviewed literature database based on the following criteria: (i) inclusion of journals from LAC (e.g. SciELO),
(ii) number of query results in English, Spanish, and Portuguese, and (iii) expert assessment of the results with a focus on
relevancy. Web of Science and Scopus databases were chosen. Results from the query were assembled into `.ref` data files.

Second, in accordance with institutional licensing, EndNote was used to retrieve documents from the assembled reference
files. This method of corpus collection showed an unequal return rate across the three languages (Table S1), which was cor-
rected for in the subsequent steps. Third, we performed a quality assessment of the corpus to assess possible bias between
query results and document retrieval within distinct sources or time periods (Year). We defined the within-corpus bias as the
difference in relative frequencies $f$ between the query and corpus such that:

$$B(\text{Year}) = \Delta[f_{\text{query}}(\text{Year}) - f_{\text{corpus}}(\text{Year})] \tag{1}$$

$$B(\text{Source}) = \Delta[f_{\text{query}}(\text{Source}) - f_{\text{corpus}}(\text{Source})] \tag{2}$$

A corpus with satisfactory quality would be one presenting minimal bias from (1-2). To address the possibility that our
automated approach could result in between-corpus bias in terms of return rate, we augmented the corpus by manually down-
loading articles to correct for biases by year and presence of Digital Object Identifier (DOI). We defined the English corpus as
well-behaved and used it as a reference to adjust the Spanish and Portuguese corpora so that each language's corpus reached
similar return rates. This adjustment reduced both within- and between-corpus bias by using systematic sampling with unequal
inclusion probabilities of the $p_i$ (Madow et al., 1949), where $p_i$ was defined using the normalized product of the bias func-
tions and $i$ refers to the $i$-th non-retrieved article. This procedure was performed twice leading to a similar return rate across
languages (Table S1).

Lastly, we performed a cleaning process to prepare the article texts for the topic model. Texts were converted into a standard-
ized version in which all cases were lowered, words not found in a dictionary were removed, and special patterns for words
such as emails and URLs were assigned tags. Additional cleaning removed punctuation (except for apostrophes and hyphens
inside words) and single-letter words. Finally, lemmatization (the reduction of a word to a common base form) was performed





using TreeTagger software by changing all nouns into singular form and all verbs into the infinite tense (Schmid, 1999). This
final version of each corpus, with lemmatization, was then used as input for topic modelling.

## 2.2  Collecting socio-hydrologic attributes

To build a database of socio-hydrologic country descriptors, we collected 43 relevant indicators from five indexes to compare
and contrast LAC countries (Table S2, Data S1). We selected the indicators from the following databases:

  – AQUASTAT, (http://www.fao.org/nr/water/aquastat/tables/index.stm)
– Environmental Performance Index (https://epi.envirocenter.yale.edu/epi-downloads)
  – Global State of Democracy (www.idea.int/gsod-indices/#/indices/world-map)
  – Social Progress Index (https://www.socialprogress.org/index/global/results)
  – LAC INFORM (http://www.inform-index.org/Subnational/LAC).

AQUASTAT is a global information system produced by the Food and Agriculture Organization of the United Nations, and
presents a perspective on agriculture and water resources availability, infrastructure to support large-scale regional planning
and analysis. The Environmental Performance Index considers two fundamental dimensions of sustainable development: envi-
ronmental health, which rises with economic growth and prosperity, and ecosystem vitality, based on 24 indicators (Wendling
et al., 2018).

The Global State of Democracy Indices depict democratic trends at the country, regional, and global levels across a broad
range of different attributes of democracy in the period 1975–2015. Democracy is conceptualized as popular control over public
decision-making and decision-makers, and equality of respect and voice between citizens in the exercise of that control. The
index translates these principles into five main democracy attributes: representative government, fundamental rights, checks
on government, impartial administration, and participatory engagement (International Institute for Democracy and Electoral
Assistance, 2017).

The Social Progress Index is a comprehensive measure of quality of life, independent of economic indicators, and is designed
to complement economic measures such as Gross Domestic Product. Social progress is defined as the capacity of a society to
meet the basic human needs of its citizens, establish the building blocks that allow citizens and communities to enhance and
sustain the quality of their lives, and create the conditions for all individuals to reach their full potential. The index aggregates
three broad dimensions of social progress: basic human needs, foundations of wellbeing, and opportunity (Fehder et al., 2018).

The LAC INFORM Risk Index simplifies risk-based information for LAC countries. A risk score is calculated for each
country by combining 82 indicators that measure three dimensions: hazard and exposure, which captures potential hazardous
events and the number of people that could be exposed; vulnerability, which measures the fragility of socio-economic systems
and the strength of communities, households and individuals to confront a crisis situation; and lack of coping capacity, which
takes into account a country's institutional and infrastructural strength to cope with and recover from crisis (INFORM, 2018).

Although these indicators do not capture the full spectrum and complexity of factors related to water, they allow for an
analysis of the topic modelling results based on country clusters with similar characteristics.



## 2.3 Electronic survey

Results from the literature review were ground-truthed with an electronic survey sent to corresponding authors of the articles from the corpus. The survey aimed to shed light on researchers' characteristics by including questions about their research experience, institutional affiliation, publication history, their perceptions on funding and interdisciplinarity in the field of water resources, and open-ended questions. The survey received exemption from the University of California, Davis IRB Administration (ID 1335782-1).

The survey was designed to include 16 questions, in three languages (English, Spanish, and Portuguese), and take approximately 5 minutes to complete (Table S3). Respondents were asked questions about their position, institutional affiliation, years of experience, main research discipline, countries of birth, residence, and research focus, number of peer-reviewed publications, motivations for picking journals for publications, source of funding, and opinions regarding interdisciplinary research.

The survey was sent in November 2018 to corresponding authors from 22,324 papers, a subset of the final corpus, using Qualtrics survey distribution software. This included articles written in each language: 20,332 English, 1,293 Spanish, and 699 Portuguese. The survey was re-sent to non-respondents weekly until February 2019, leading to 1,969 responses. The survey response data was cleaned and prepared for analysis. The survey responses in Spanish and Portuguese were translated to English and compiled into one document.

## 3 Methods

In this section, we detail: (i) the clustering of countries from their socio-hydrologic data; (ii) the generation and mining of metadata; and (iii) the metrics to identify bright and blind spots of water research.

### 3.1 Clustering analysis

We clustered countries based on socio-hydrologic variables using two different methods: k-means clustering (Hartigan and Wong, 1979) and hierarchical clustering (Murtagh, 1983). We performed the clustering with Euclidean distances and following Ward's criterion. We then investigated the optimal number of clusters by evaluating the evolution with the number of clusters of the total within sums of squares and of the average silhouette width (Rousseeuw, 1987). In addition, we used the following four validation metrics to assess the stability of the clustering under the complete set of clustering variables and performed an iterative procedure where one variable is removed from the set, an approach akin to leave-one-out cross-validation:

- the average proportion (APN) measures the proportion of observations not placed in the same cluster under both cases and evaluates how robust the clusters are under cross-validation (Datta and Datta, 2003);
- the average distance between means (ADM) measures the variation of the cluster center and evaluates the stability of the localization of the cluster in the multi-dimensional clustering variable space (Datta and Datta, 2003);
- the average distance (AD) measures the distance between observations placed in the same cluster and evaluates within-cluster stability (Datta and Datta, 2003);





– the figure of merit (FOM) estimates the predictive power of the clustering algorithm by measuring the within-cluster
variance of the removed variable (Yeung et al., 2001).

## 3.2 Metadata generation and mining

Each document in the corpus was augmented by generating and mining metadata. The mined metadata correspond to author
keywords as well as the citing and cited literature resulting in a citation network. The generated metadata correspond to
modelled topics and study location.

### 3.2.1 Topic model

The content of the corpus documents were modelled using Latent Dirichlet Allocation (LDA), a Bayesian, generative, proba-
bilistic model conceptualizing each document in a corpus of documents as a random mixture of topics (Blei et al., 2003). Topics
are Bayesian bag-of-words corresponding to a distribution over the vocabulary (i.e. the words appearing within the corpus). In
essence, one topic corresponds to words that have a significant probability to co-occur.

The LDA was programmed to identify 105 topics in English and 65 topics in the Spanish and Portuguese corpora based on
commonly used metrics for LDA tuning. We then conducted a quality assessment of the topic models through cross-validation.
For this we developed human-derived topics for the English Corpus by reading a subset of 1,428 papers from the corpus and
manually identifying single-word tags based on keywords and main research topics. A similar percentage of documents were
read for the Spanish and Portuguese corpora: 188 and 111, respectively.

As topics are statistical objects, they must be assigned a human label to make them tractable. An interdisciplinary review
panel of eight water experts therefore assigned labels for each topic while simultaneously evaluating their significance. Topics
were removed if multiple members of the review panel determined that the nost frequently occurring words were irrelevant
based on their expert knowledge of water resources science. The remaining relevant topics were tagged with five labels inde-
pendently by multiple reviewers and the collection of proposed topic labels were then harmonized to produce final topic labels.
Topics were assigned labels for each of several levels: (i) specific topic name; (ii) theme; categories of scientific research as
defined by the US National Science Foundation (NSF) that were either (iii) specific, or (iv) broad; and v) description: spatial
scale, water budget, or methods. These labels were consolidated into four topic categories: general, specific, methods, and
water budget.

### 3.2.2 Metadata mining and citation network

Article metadata containing the citing literature and author-defined keywords for articles from the English corpus was extracted
using Elsevier API and each article's DOI. We used this citing literature to build a citation network. Keywords were used to
supplement the country location labels assigned during human-reading by looking for regular expressions of the name of the
target countries. Of the 31 countries in LAC, only 23 countries had a sufficient occurrence (i.e., at least 30 articles) within the
human-read subset of the English corpus and were included in the citation network.





The citation network was extracted from article metadata, which stores identifiers for the citing article (i.e. the article to
which the metadata are attached to), and the cited articles. A total of 29,900 citations were found between 4,603 unique articles
of the English corpus. The resulting bibliographic network is defined by its $4,603 \times 4,603$-adjacency matrix $B$ and was filtered
to remove edges between nodes external to the collected corpus. The citation network between countries, $B_C$ was obtained by:

$$B_C = C^T \cdot B \cdot C \tag{3}$$

where $C$ is a $4,603 \times 23$-matrix containing the output of the machine learning probabilistic predictions of the location of
study between 23 possible countries of study (see below). In consequence, $B_C$ is a weighted directed adjacency matrix. Similar
reasoning leads to the citation network between specific research, method, and water budget topics by changing the matrix $C$
in the previous equation for the LDA probabilistic predictions of topics.

### 3.2.3 Location prediction

We used machine learning to predict the location of the country of study of each paper in the English corpus. The training labels
were provided by human-reading randomly chosen articles from the corpus (1,428 human-derived labels) and from text mining
the article metadata (2,663 text-mined labels). Interestingly, the human-reading provided 563 observations of irrelevant country
locations (i.e. outside LAC) or irrelevant subjects of study (i.e. not water resources related). This occurred in some cases when
our queries returned articles containing accurate keywords but different meanings than intended; for example, the search work
"Mexico" returned irrelevant locations included regions of the United States around the Gulf of Mexico, and irrelevant topics
such as signal processing analyses employing the "Mexican hat" wavelet.

The human-derived labels were first used for constructing a relevance filter based on simple binary classification between
"Relevant" and "Irrelevant" documents ($n = 1,386$ after quality control). The predictors consist of the text document term
matrix derived from the cleaned corpus text using tokens related to country names, and of the topic membership output by the
topic model ($p = 138$). A benchmark of the six following models was conducted: featureless (baseline), random forest, support
vector machine, naive Bayes, multinomial regression and extreme gradient boosting. The hyper-parameters were initially set
at standard default value. The resampling scheme was 10 repetitions of 10-fold cross-validation. Best performing models
were selected for further tuning using a nested cross-validation (Bischl et al., 2012) with a simple hold-out inner loop and 10
repeats of 10-fold cross-validation as an outer loop. The size of the tuning grid was set to 16 between standard values for each
hyper-parameter.

The prediction of the location of study for each document was performed using both human-derived and text-mined labels
($n = 3,494$ after quality control). The predictors consisted of the text document-term matrix derived from the cleaned corpus
text using tokens related to country names ($p = 33$). Models and resampling schemes were similar to the ones used for the
relevance filter. The hyper-parameters were initially set at standard default value.





### 3.3 Bright spots and blind spots

225 This section details assessments of: (i) ground-truth from analyzing survey data; (ii) research volume; (iii) research spread using topic normality; and (iv) research connectivity from network analysis.

#### 3.3.1 Survey analysis

Close-ended responses of the electronic survey were analyzed by tallying aggregate data by country and discipline of study. This allowed for a number of inferences such as the most commonly represented research disciplines and the countries of study 230 and of origin of respondents. Research collaborations were analyzed based on the three main countries of study and the three main countries of research collaborations for every respondent.

Open-ended responses were coded for content and analyzed in Atlast.ti, resulting in dozens of codes used to group responses of similar content (Table S7). Comments irrelevant to the study were removed. We also identified relationships between codes based on connections in the data. If respondents mentioned political issues that hamper funding availability, we coded the two 235 elements (e.g. "political issues" and "funding difficulties", and then linked them with the qualifier "[...] is the cause of [...]"). In addition, through the 'word cruncher' tool, we generated a word frequency from the survey comments resulting in a word cloud visualization where words' sizes are proportional to their use frequency.

#### 3.3.2 Research volume over time

A timeline was created with the number of new research articles published per year representing countries in the three socio-240 hydrologic clusters, to visualize growth in research output over time. The articles were sourced from the English corpus of water resources research, because these articles were labeled with the country of research and could therefore be associated with a socio-hydrologic cluster. To better understand trends observed in each socio-hydrologic cluster, a residual analysis was performed. The data were transformed with a logarithmic transformation to obtain a roughly linear relationship between time and research output, and then a linear regression was calculated. The residuals for each year were plotted and displayed starting 245 from 2000 to 2017. Year 2000 was chosen as the starting point because it marks the time by which research output had increased enough to reach at least 30 new articles in each socio-hydrologic cluster per year. The residuals were then plotted along with brackets of the standard deviation (for both positive and negative values) to provide a reference of significance.

#### 3.3.3 Topic normality

The normality of research topics was estimated for general and specific topics, as well as for the method and water budget 250 topics. Documents in each subset were sourced from the English corpus. Each subset was filtered for documents that were labeled with a country of research, and then for countries where the sum of documents per country was greater than 30. A statistical distance from standard, normal distribution was calculated to describe the normality of topic probabilities from two perspectives: across documents and across countries. As a statistical distance, we chose the Jensen-Shannon distance because of its link with entropy (Lin, 1991) and its interpretation in terms of a proper distance (Endres and Schindelin, 2003).





The Jensen-Shannon distance, $d_{JS}$, is calculated between two probability density functions $N$ and $P$ and $d_{JS} = 0$ if $P = N$ (see Appendix A for full derivation). Here, we chose $N$ as the standard normal distribution, $\mathcal{N}(0, 1)$. $P$ is the standardized probability distribution of topic probability distribution across either documents or countries. To better meet the intuition of normality, we define it as $1 - d_{JS}$ so that normality is closer to 1 when the $P$ is closer to the standard normal distribution.

### 3.3.4 Citation network analysis

The citation network was analyzed using Gephi 0.9.2 (Bastian et al., 2009). Descriptive parameters and geometry metrics were calculated. We conducted this network analysis at each LAC country level for general and specific topics, and water budget and methods topics. Probability adjacency matrices were extracted from predictive model results and normalized to highlight the relationship between topics or countries regardless of sample-size in our corpus.

A force-directed graph algorithm, Fruchterman-Reignold (FR), was selected, to produce the networks' visualizations. It
simulates the graph as a system of mass particles. The nodes are the mass particles and the edges are springs between the particles. The algorithm tries to minimize the energy of this physical system. FR is most suitable for small networks and a better performance (Fruchterman and Reingold, 1991; Jacomy et al., 2014). The force directed citation networks show the degree of connectivity by the nodes' size and edge thickness, but there is no distinction between positive or negative citations (Bruggeman et al., 2012).

For each network, the following geometric descriptions were calculated: number of nodes (countries or research topics), number of edges (citation between countries or research topics) and thickness of the edges (connectivity proportion). We also calculated network density, and degree. Network density is a measure of the connectedness of a graph, defined as the number of connections, divided by the number of possible connections; with all possible edges and density = 1 (Tonin et al., 2019). Degree is the number of connections each node (country or topic) has with another node (country or topic). Degree has
generally been extended to the sum of weights when analyzing weighted networks and labelled node strength, so the weighted degree and the weighted in- and out-degree was calculated (Newman, 2001; Barrat et al., 2004; Opsahl et al., 2010). In all visualizations, edge thickness represents weighted degree and network node size represents research volume defined as the sum of probabilities for the given node in the corpus.

## 4 Results

### 4.1 Clustering results

Both clustering methods yielded similar results. The total within-sum of squares evolved after two clusters are chosen. Similarly, the average silhouette width strongly exhibited a peak for two clusters. Further inspection of clustering in principal component dimensions indicated that the cluster with Mexico and Brazil was significantly distinct from all other countries, explaining the observation of a sharp peak in average silhouette width. However, validation metrics exhibited optimal null





values of APN and ADM for two or three clusters. In addition, AD and FOM were lower for three clusters than for two. Based on these results, we chose three clusters to describe the grouping of countries based on their socio-hydrologic variables.

## 4.2  Metadata generation

### 4.2.1  Topic model

We determined if the LDA successfully identified a relevant topic based on the top 10 occurring words which showed a 86%
agreement between expert-identified topics and LDA-derived topics. We judged the performance of the LDA by comparing the topic model output from the English corpus to the output from the Spanish and Portuguese corpora. The specific topic label for each topic was used for comparison. The number of topics with each specific label were grouped by language and tallied (Fig. 1, Tables S4-6).

Our findings are based on the output of a topic model of articles written in English and are predicated on the assumption
that the English language corpus accurately represents the breadth of regional research published in English, Spanish, and Portuguese given that non-English corpora were small fractions of the English corpus (4% and 2% for Spanish and Portuguese, respectively). Comparing the topic model performance of the three corpora (Fig. 1) supports this assumption. One-third of the specific topics present in the English corpus are present in all languages. Importantly, this subset of topics includes the majority (10 of 16) of the top 25% of research topics from the English corpus. Another third of the topics are present in two of the three
languages and the last third is only present in English. Because of the small size of the Spanish and Portuguese corpora, it is likely that their topics do not cover the entire scope of peer-reviewed water resources research published in these languages. While access to more articles would have resulted in even closer alignment with the English corpus, the existing alignment between topics present across all three corpora is sufficient to support our decision to base topic model conclusions on the English corpus alone.

### 4.2.2  Relevance filter and location prediction

For the relevance filter, the random forest, multinomial, and support vector machine models were the best-performing models and showed no statistical difference in the distribution of their performance measured by area-under-curve (AUC, Fig. 2a). Random forest and multinomial were the best-performing tuned models and showed no statistical difference in the distribution of their performance measured by AUC (Fig. 2b). The distribution of the hyper-parameter resulting from the nested cross-
validation showed a better constraint for the multinomial model which was selected for predictions. 12,616 articles were predicted as relevant to our study. The performance of this multinomial model corresponds to a mean AUC of 0.82, a mean accuracy of 77% and a true positive rate of 86%.

For predicting location, random forest outperformed every other model with a mean multiclass AUC of 0.99 and a mean accuracy of 96% (Fig. 3). No further tuning was performed and the random forest with default hyper-parameters was used
for predictions. More complex approaches were investigated: using additional geographical tokens (e.g. rivers and mountain ranges names), multilabel classifications (e.g. binary relevance, label powerset), or deep learning models of natural language







**Figure 1.** Topic model performance assessed by the total number of specific topics present in each language. Bolded topics represent the top 12.5% of research (named topics on 6). Numbers in each box correspond to the number of topics from the topic model that were manually grouped together under a common label.





**Figure 2.** Machine learning performance for relevance prediction based on simple binary classification between "Relevant" and "Irrelevant" documents ($n = 1,386$). a) Performance of untuned models; b) Performance of selected tuned models. Brackets highlight statistical differences between distributions.





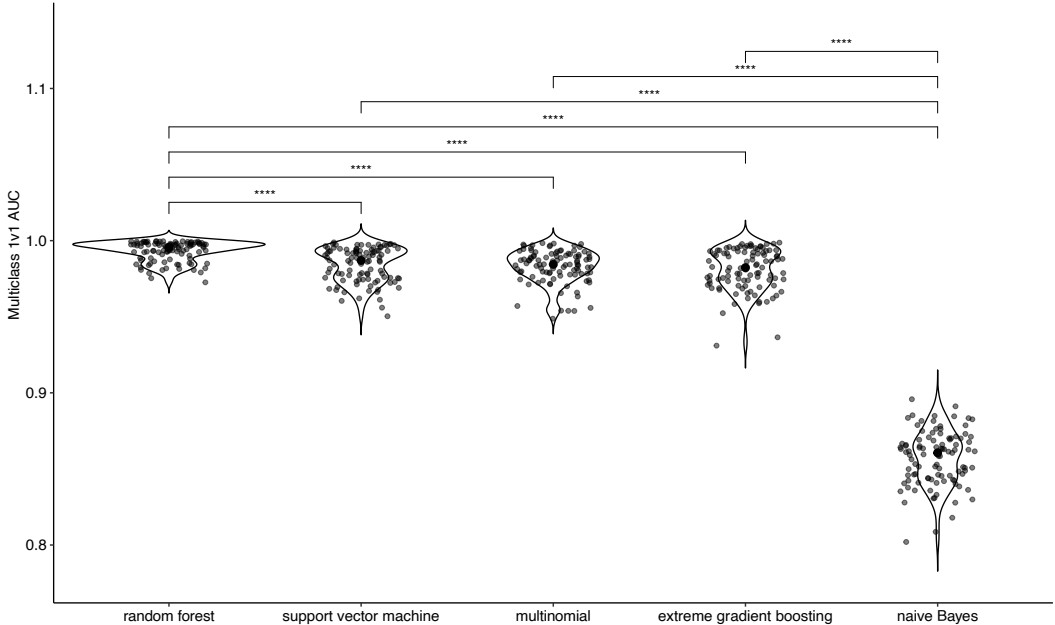

**Figure 3.** Machine learning performance for prediction of the location of study for each document using both human-derived and text-mined labels ($n = 3,494$). Brackets highlight statistical differences between distributions.

processing (e.g. Google's BERT, Devlin et al., 2018) both on full texts and abstracts. These more complex methods yielded similar results than our simpler initial approach and were therefore not pursued further for this study.

## 4.3 Survey results

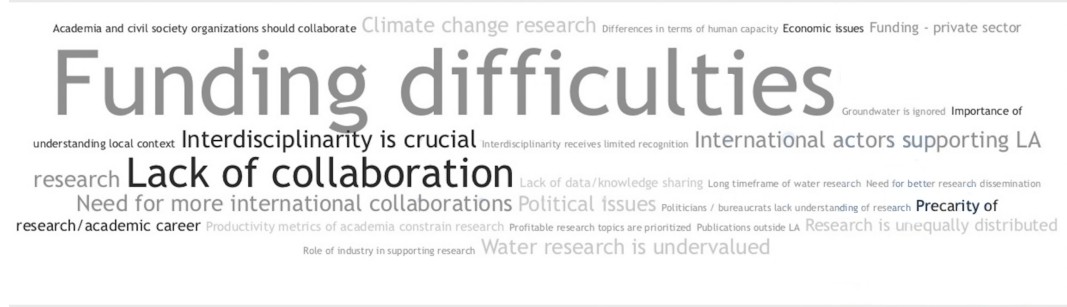

**Figure 4.** Word cloud based on word frequency from survey comments.

The most common categories of answers in the comment section were summarized (Table S7). Difficulties related to funding stood out as the main challenge for respondents (Fig. 4). These were often linked to political or economic trends in their countries. Respondents further explained that available funds are not only distributed unequally across countries, but also

within countries, and across research areas. Collaboration, or the lack thereof, was another important theme. Many respondents claimed that they collaborated more with North America, Europe, and Australia than with other LAC countries, in large part
due to funding opportunities. Many indicated an interest in collaborating with researchers in other LAC countries.

## 5 Bright spots and blind spots

### 5.1 Volume: Growth through time

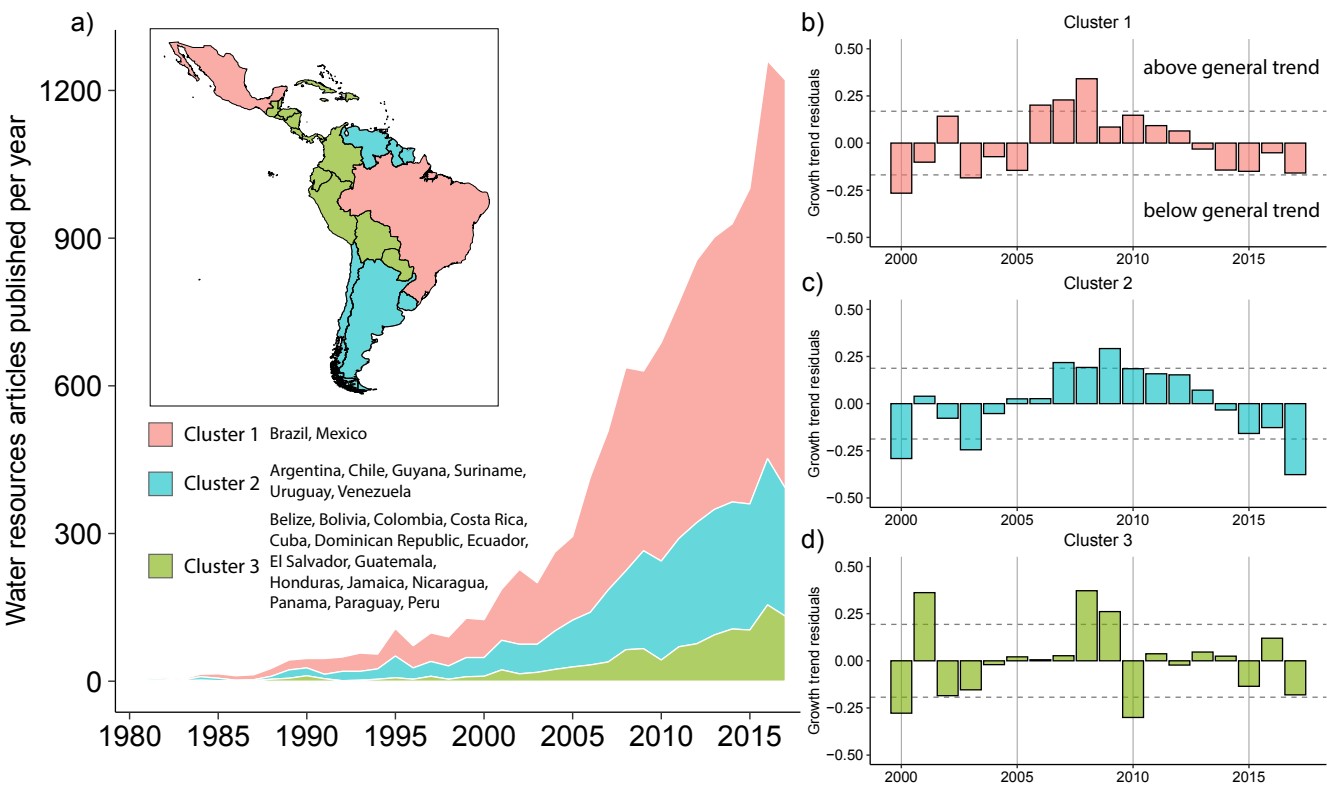

**Figure 5.** Growth of water resources research. a) New articles per year for each socio-hydrologic cluster (inset): b-d) Residual analysis estimating the deviation from the exponential trend for each socio-hydrologic cluster.

First, we look at research *volume* by country and topic. The cumulative research growth showed an exponential growth trend over the last four decades, although the trend was more evident in Cluster 1 (Brazil and Mexico) than in Clusters 2 or 3 (Fig.
5). These results were used to interpret periods of time in which the scientific production of water research was increasing or decreasing on a region-wide scale, in reference to the growth trajectories of each socio-hydrologic cluster. Reasons for this growth could relate to a wide array of changes in LAC over the last forty years beyond an increasing interest in addressing





water-related knowledge gaps: increased access to the internet and online resources, availability of existing research in online repositories, and expansion of opportunities in higher education (Delgado-Troncoso and Fischman, 2014; Baskaran, 2017).

A residual analysis identifies three distinct periods since 2000, the first year that each socio-hydrologic cluster was represented by over 30 research papers in each language. Annual output was lower than each cluster's general trend for the first several years, followed by a period of relatively higher output from 2007-2012, ending with a trend of decreasing growth from 2013-2017, although some residuals are below a single standard deviation within these periods. It is possible that uniform anomalies below or above general growth trends correspond to region-wide events, although a several-year lag could be pos-

sible between causal events and effects in research output. For example, a connection may exist between Brazil's economic crisis starting in 2012 and the subsequent drop in research output from 2013-2017.

## 5.2   Where and what is being studied?

Combining research topics with predicted study location describes the composition of water research in LAC with a chord diagram (Fig. 6). This chord diagram was obtained from the weighted bi-partite network between countries and topics (Fig. 6).

For legibility, the network was filtered to remove the edges with weight lower than the 75$^{th}$ percentile of the edges of a given country. In other words, for each country, only the top 25% links are displayed.

   While research about Brazil, Mexico, Argentina, and Chile are bright spots that dominate the research landscape, the absence of countries in the Caribbean and most of Central America indicates a shortage of research in these regions. A country's socio-hydrologic cluster correlates to its representation in overall research, suggesting that, more than population size alone, a

country's water and economic resources, geography, and history influence the likelihood that researchers study that country.

   Mexico and Argentina alternate for second highest representation, depending on the topic, after Brazil. These findings are confirmed by survey respondents, of which 35% study Brazil, followed by Mexico with 15% and Argentina with 9% (across all research topics). Only 12% of survey respondents focused most of their research on countries in the Caribbean or Central America.

Water research is not distributed equally among disciplines and is primarily conducted in the physical and life sciences, representing together 80% of topic probabilities. Survey responses also confirm these findings, as 80% of respondents identified as life and physical scientists. The lack of water research in social sciences may reflect a combination of low publication rates (compared to physical sciences), a historical framing of water management as a purely technical discipline (Callaghan et al., 2020), and financial resources from governments biased towards physical sciences and engineering.

## 5.3   Spread: research on reservoirs and risk assessment is siloed

Next, we look at the *spread* of water research in LAC, described by topic normality for general, specific, water budget, and method topics. Importantly, Caribbean nations and most of Central America did not have enough research to be included in these analyses and represent blind spots. For general topics (Fig. 7a), physical and life sciences represent bright spots, with topics spread normally across both countries and documents. Conversely, underlining findings from research volume (Fig.

7), the limited spread of social sciences across documents represents a blind spot. For specific topics (Fig. 7b), hydrology,



**Figure 6.** Composition of water research in LAC according to study location and top 25% of studied research topics. On the left, countries are identified individually and by their associated socio-hydrologic cluster. On the right, research topics are grouped by their general category. The top 50% of specific topics are listed within each general category.





**Figure 7.** Normality of research topics for a) general topics and b) specific topics. Inset shows the countries excluded from the analysis (gray).





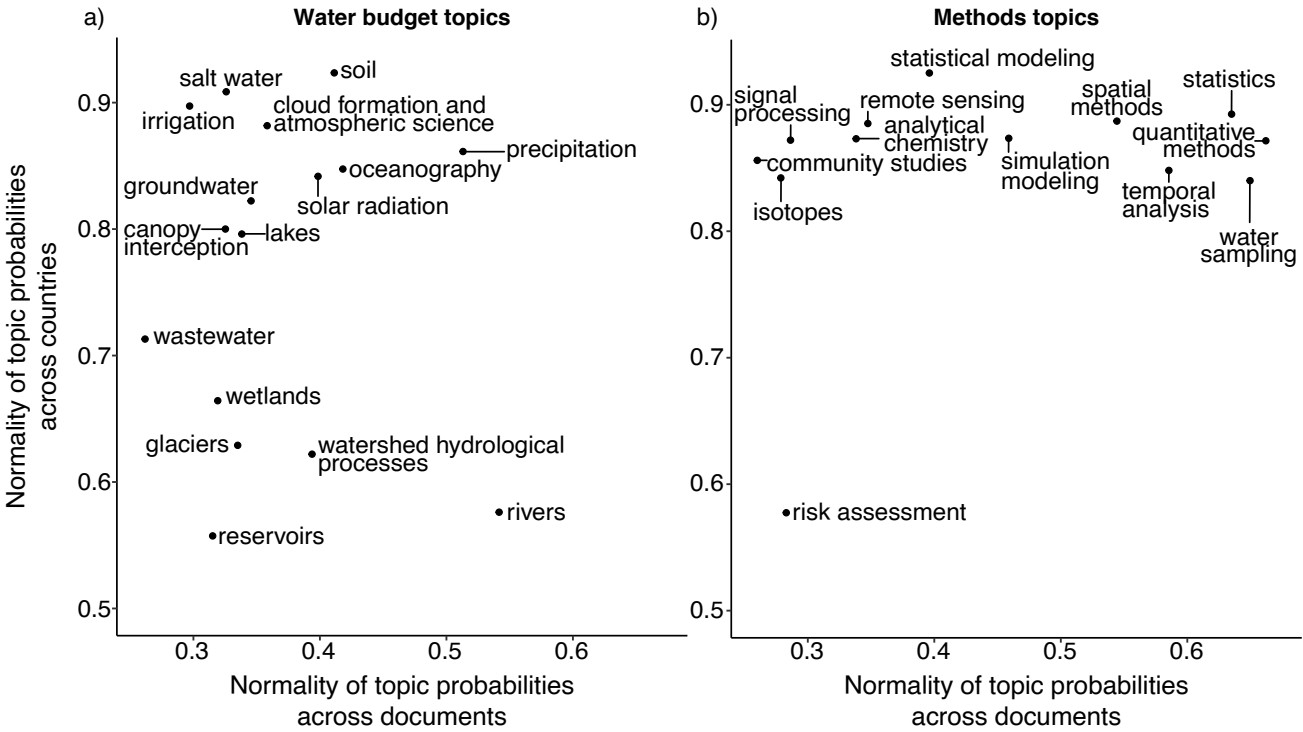

**Figure 8.** Normality of research topics for a) water budget topics and b) methods topics.

atmospheric sciences, statistics, and mathematics are bright spots across countries and documents, while marine biology and forest science are likely blind spots. The majority of specific topics have high normality across countries ($> 0.8$) but low normality across documents ($< 0.4$). These topics are either infrequently mentioned or, if mentioned, are the sole subject of a paper, lacking integration in interdisciplinary research. For water budget topics (Fig. 8a), precipitation, which must be

monitored and understood to manage water resources, has a distribution closest to normal, representing a bright spot, while glaciers is far from having a normal distribution, likely because few countries have glaciers to study (i.e. Argentina, Bolivia, Chile) and they are relatively less relevant for other countries' water budgets. For method topics (Fig. 8b), statistics, quantitative methods, and water sampling are identified as bright spots of water research methods across the region.

The least normality is seen in two topics of great importance for water management: reservoirs and risk assessment. Each
topic has normality far from 1: less than 0.6 across countries and less than 0.3 across documents (Fig. 8). These are alarming blind spots given the importance of reservoirs for water supply reliability, their impacts on local communities and ecosystems, and potential consequences associated with a lack of planning and risk management as climate change increasingly acts as a risk multiplier, including for reservoir operation. Although it is possible that information on reservoirs and risk assessment exists in grey literature, such as government reports and university publications, this information is not as accessible as scientific
publications and limits regional knowledge-sharing.





## 5.4 Connectivity: *Brasil é nùmero um*

We complete our description of LAC's water research portfolio by estimating degrees of *connectivity*. Four network graphs were produced. The network graphs showed a density value of 1; as expected, all topics and countries are connected. The country citation network identified 23 nodes and 529 edges. The network degree of connectivity has a maximum degree

(<50%), medium degree (<30%) and minimum degree less than (<10%), where each node corresponds to a LAC country, and colors represent socio-hydrological clusters. The general topics network identified 5 nodes and 25 edges, with a degree of connectivity from a maximum of 45% to a minimum of 27%. The specific topics network has 43 nodes and 1,849 edges, with a maximum degree of connectivity of 18%, a medium degree of 10% and a minimum of 5%. The water budget and topics network has 27 nodes and 729 edges, and a maximum degree of connectivity of 5%.

The high volume of research about Brazil (45% of the labeled English corpus) motivated further investigation to see if this large scientific output is proportionally more influential than the research about other countries. Brazil is a central bright spot of the citation network (Fig. 9), as citations by publications from all countries, including Brazil itself, are primarily directed toward research about Brazil (49%). Furthermore, the geographic location or socio-hydrologic cluster of a country do not appear to indicate the extent of citations of researchers working in that country. For example, although Argentina and Chile

share a particularly long border and a rich history of political and economic ties, the citation network reveals only a medium degree (14%) of connectivity between the two countries. Similarly, countries in the Caribbean and Central American regions that share many similarities, including water management challenges, show a low degree of connectivity (less than 5%).

Results from our survey complement findings from the citation network. Over half of participating researchers collaborated with researchers outside of the country(ies) they study. Despite being the only Portuguese-speaking country, Brazil was

the country most often listed in collaborations within LAC (17%). A quarter of research collaborations involve non-LAC researchers, mostly in the United States (14%). This may reflect differences in access to funds and highlights how more affluent countries can influence the scope of research conducted in LAC. Respondents indicated that insufficient and precarious funding arrangements are their main challenge. 89% of respondents said that the government is their main source of funding, which may explain that a country's political and economic context were mentioned as further aggravating funding availability. Fund-

ing difficulties were also associated with a lack of value given to water research and to the long timeframes associated with research that are misaligned with decision-makers' timelines.

Furthermore, being physically close or part of the same socio-hydrologic clusters did not increase the likelihood of cross-country collaboration between countries. For instance, 22% of researchers in the neighboring countries of Argentina or Chile either study both countries or collaborate with one another, while 24% of those researchers report collaboration with Brazil.

Researchers from Mexico and Brazil, who share a socio-hydrologic cluster, collaborate even less, with only 14% reporting to work or collaborate in both countries, despite a high level of connectivity from Mexico to Brazil in the citation network. Conversely, more than 80% of researchers in the Caribbean reported collaborating with researchers from other Caribbean nations and few collaborated with Brazil. This is in opposition to the findings from the citation network showing few citations





**Figure 9.** Connectivity between LAC countries, measured by directional citations between articles' country of study. The direction of each edge is represented by drawing it clockwise from an earlier node to a later node.





within the Caribbean and more frequent citations of publications on Brazil, but this could partially be due to the limited number

of articles studying the Caribbean region included in our corpus.

## 5.5   Opportunities exist for building research communities

**Figure 10.** Connectivity between topics of research, measured by directional citations between articles' research topic for a) general topics and b)specific topics. The direction of each edge is represented by drawing it clockwise from an earlier node to a later node.

We assessed the connectivity of water research throughout the region by aggregating research from all countries by topic (Fig. 10). A reciprocal relationship between the physical and life sciences dominates the citation network, while research within social sciences, engineering, and mathematics is cited less often (Fig. 10a). Research connectivity from all sub-disciplines is

heavily polarized towards research on hydrology and water resources, however certain topics display a comparable degree of





self-citation (Fig. 10b). This may indicate that water resources research communities in political science and governance, civil engineering, ecology, geophysics, and toxicology are siloed or more connected to non-water resources research communities.

Interestingly, behind these few siloes and the central node, a vast network of connectivity exists. While this level of connectivity is low (less than 10%), it characterizes water resources as a scientific discipline where research topics are already
integrated, albeit with room for strengthened interdisciplinarity. Survey respondents confirm this low level of background connectivity, reporting dissatisfaction with the interdisciplinarity of their research. Nonetheless, an opportunity exists to build off these existing connections to form communities of researchers and strengthen research impact through knowledge sharing and collaboration across disciplines (Uzzi et al., 2013; Astudillo, 2016; Larivière et al., 2015).

### 5.6 Limitations and future research opportunities

The wide scope of this study, intended to capture the breadth of the state of water resources research across LAC, required inevitable compromises in the depth of information and the subsequent ability to thoroughly interpret our results. Notably, much scientific literature in Spanish and Portuguese was not readily available or accessible online, and resulted in the need to rely on English publications as a proxy of research across LAC. A targeted method to collect gray literature would increase the size of the Spanish and Portuguese corpora. Of the literature we found, very little focused on Caribbean countries, and this lack
of information limited subsequent analysis. A targeted method of corpus augmentation and human reading validation towards less-represented countries and topics will likely increase the model's predictive capabilities and may improve the representation of Caribbean countries.

In addition, while the presented citation network included all LAC countries, the exclusion of countries outside of LAC prevented a more comprehensive analysis of LAC countries' reliance on non-regional research. Survey responses suggested
that reliance on non-LAC research was high, as researchers stated they were more likely to collaborate with scientists outside of LAC than within LAC. Inclusion of non-LAC countries in the analysis of scientific interactions could present many opportunities for expanding our findings in future research. Finally, our study indicated where bright and blind spots appear across research in LAC, but did not aim to examine causal relations for these patterns, a common shortcoming in Science of Science (Fortunato et al., 2018). Survey results indicate that funding for research is an important driver influencing where bright and
blind spots occur.

However, a more comprehensive answer would require exploring historical, political, economic, and social dynamics influencing the allocation of research resources. Overall, this work displays the value of our novel method to interpret results from machine learning, points to the need for a deeper and wider understanding of existing water resources research in water vulnerable regions, and warrants expanding our methods to include gray literature and coverage across the Global South.

## 6 Conclusions

This unprecedented multilingual literature review provides insights into bright and blind spots of water research throughout LAC. Our results reveal that the region's vulnerability to water-related stresses, and drivers such as climate change, is com-





pounded by research blind spots in certain topics (e.g. reservoirs and risk assessment) and in entire sub-regions (e.g. Caribbean nations). Although certain topics and countries are under-studied in relation to the rest of the corpus, research on most com-
ponents of the water budget (e.g. precipitation) represents a bright spot and suggests that most countries can make science-informed decisions regarding their water management. Research on water resources in Brazil dominates the research landscape, representing another bright spot. However, Brazil's dominance also highlights a regional vulnerability: while research on Brazil is vast, well-rounded, and highly influential across LAC, funding cuts and policy shifts that affect the country's scientific output can halt progress and impede scientifically-informed water management throughout the region. Supporting societal and ecolog-
ical needs while addressing challenges linked with future water-related risks will depend on countries' abilities to improve the accessibility of existing research (in English, Spanish and Portuguese), expand research in under-studied topics (particularly in the social sciences), and harness existing opportunities for knowledge sharing.

*Code and data availability.* Code and data are available as the R package `wateReview v0.1` archived at: http://doi.org/10.5281/zenodo.4552771. Long form documation is available at: https://hrvg.github.io/wateReview.

**Appendix A: Jensen-Shannon distance derivation**

The Kullback-Leibler divergence measures the expected information for discriminating between discrete probability distributions $P$ and $Q$ when only observing $P$ (Kullback and Leibler, 1951):

$$D_{KL}(P,Q) = \sum_{i=1}^{n} P(x_i) log_b \frac{P(x_i)}{Q(x_i)} \tag{A1}$$

The Kullback-Leibler divergence is *asymmetric* and, in non-trivial cases, $D_{KL}(P,Q) \neq D_{KL}(Q,P)$. The Jensen-Shannon
divergence is a *symmetric* measure of discrimination between the two probability distribution functions $P$ and $Q$. It is directly related to the Kullback-Leibler divergence (Lin, 1991; Topsoe, 2000):

$$D_{JS}(P,Q) = \frac{1}{2}[D_{KL}(P,R) + D_{KL}(Q,R)] \tag{A2}$$

with $R = \frac{1}{2}(P+Q)$ the midpoint probability distribution.

Finally, the Jensen-Shannon distance, $d_{JS} = D_{JS}^{1/2}$ retains the advantageous symmetric property of the Jensen-Shannon
divergence, but also satisfies the triangular inequality making it a proper distance metric (Endres and Schindelin, 2003). This property allows for the construction of distance matrices; a common tool in data analysis (e.g. correlation matrix).

*Author contributions.* A.J.D and H.G. contributed with Conceptualization, Methodology, Investigation, Data management, Analysis, Visualization, Project administration, Funding acquisition, and Writing - original draft, revisions & editing. R.D.G., N.P.K., and F.v.d.B. contributed



with Conceptualization, Methodology, Investigation, Data management, Analysis, Visualization, and Writing - original draft, revisions &
editing. A.K. contributed with Methodology, Investigation, Data management, and Writing - revisions & editing. J.P.O.P. contributed with
Conceptualization, Investigation, Analysis, Visualization, and Writing - revisions & editing. L.E.G.D contributed with Conceptualization,
Investigation and Writing - original draft, revisions & editing. J.G.R. contributed with Investigation. E.G. contributed with Investigation and
Writing - revision & editing. S.S.S. contributed with Conceptualization, Methodology, Investigation, Project administration, and Writing -
revisions & editing.

*Competing interests.* The authors declare no competing interest.

*Acknowledgements.* We are indebted to Shanti Sandosham, Miranda Romero, Lilly Mccaffrey Pecher, interns at the University of California,
Davis, (UCD) and Carly M. Lawyer, intern at the University of South Carolina. Pamela Reynolds and Carl Stahmer at the DataLab (UCD)
provided invaluable insights. This study was supported by funds from Jastro Research Fellowship (UCD).





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
