# Peer review of "Bright and Blind Spots of Water Research in Latin America and the Caribbean"

_Hydrology and Earth System Sciences, 2021_

## Community Comment (CC2)

**REVIEW OF THE BRIGHT AND BLIND SPOTS OF WATER RESEARCH IN LATIN AMERCA AND THE CARIBBEAN**

By Daniel Prieto Garra[1]

Very valuable and excellent work!!!. It added both very valuable innovative methodology for meta-analysis works and a great information from LAC research works on water resources.

Of course, I have nothing to add to your excellent innovative methodology. I am just adding some comments from my point of views that can help the discussion of your findings and that I hope could be useful for the participatory review of your article (a review process that I celebrate and congratulate you).

**Introduction**

I agree with most of your statements in this section **since** they are close of an objective characterization of the water resources and its problems in LAC

**Line 20 and the following**. Just as simple information, I would like to highlight than Argentina, like Brazil and Chile, is another clear example of the uneven distribution of the Water Resources in the continent. Water rich areas, are only 24% of the country territory, own 82% of the surface water resources ...Arid and semi-arid regions, the complement 76 % of the country area has only has 18% of the surface water resource and 80% of them are in the Patagonia. In Argentinean case, 72% of its population live in rich water areas where the population density reach 21 inhabitants/km$^2$, while in the semi-arid and arid region it is only 5.9 inhabitants/km$^2$.

The above does not mean that Argentina has water resources problems only in the semi and arid region. Rich water areas suffer periodic floods that affect mainly more poor people that live in vulnerable areas of the big cities and also suffers of frequent dry short period during the summer that affect mainly the agricultural production the principal base of Argentinean economy. Water availability and water quality are of course the main water related problem in the Arid and Semi-arid regions. Natural derived water resources problems are amplified in both area by a lack of a professional water management due financial constraints and institutional and human resources weakness.

**Line 33.** "Water management is a relatively young field of study and suffers from a lack or interdisciplinary and integrative perspective………" Crucial statement from my point of view.

**Line 59.** How do you explain the low number of response to your invitation to corresponding authors?

**Subsection 4.2.2**

**Line 320 to 325**. In my opinion this lines pointed out one of the main problems of the LAC´s researchers. The unequal distribution of funds within the country (always large research group of the rich areas of the country access to fund, even those competitive ones) and the lack of

[1] Dr (Wageningen, The Netherlands), MSc (Wageningen, The Netherlands) Agricultural Engineering (UdelaR, Uruguay). Former National Coordinator of the Water Resources Research Program (2001-2018) of the National Institute for Agricultural Technology (INTA) from Argentina. Freelance consultant (2019-2021).

continuity of the political and economic context that affect research support policies in our countries.  Due these points, LAC researcher strategies focus on look for collaboration with researchers from USA and especially from Europe, (more funds can be find in this area). I would add that there is a lack of incentive from our governments to support and strength LAC´s research network. I limited positive example that I know, was the strategy for foment regional research network of the PROCISUR (a cooperative program of the agricultural research institute from Argentina, Brazil, Chile, Paraguay and Uruguay) during the first decade of 2000´s. However as soon the institutes had financial problem related with neo-liberal policies implement by the government of the region it was discontinued.

**Lines 357 to 359**. I agree with your finding about water research in social sciences based on the meta review of peer-reviewed literature. Nevertheless I consider that your focus on peer-reviewed literature bias your findings. From my personal experience, I have the qualitative idea that research on socio-technical problems on water management in LA has increased sharply during the last decade. Findings of this research (mainly study cases and/or participative research with local actors) are seldom submitted to peer-reviewed journals. They are mainly reported in books (see Boelens et al, 1998, 2002, 2005), regional congress, or directly translated to diffusion publications.

From the above I would recommended you to explore this type of publication in a future update of your brilliant, useful and great works.

---

## Author Comment (AC1)

**Authors response**

We sincerely thank the reviewers and the community members for their time and effort to provide thoughtful comments on the manuscript. The positive responses are very much appreciated. We will do our best to address any suggested changes in a revised manuscript. Our proposed changes are explained in our detailed responses in the following pages (in blue), which are outlined in the following table of contents. The final section labeled 'Additional considerations' includes suggested changes the authors would like to mention for the editor's consideration.

**RC1:  José Luis Arumí, 20 Apr 2021**

The article is very interesting and as another commenter said, easy to read. The impressive number of articles analyzed and the methodology explained allows the reader to have a synthesis of the water research in Latin America and the Caribbean (LAC)

Regarding figures 5 and 9, it is clear that the larger number of publications are produce in Brazil and Mexico. However, those countries are also the LAC countries which more inhabitants, therefore, it would be interesting to see the same result normalized by the population of each country

> Authors response: This is an interesting comment which we considered early on and led us to design and implement the socio-hydrologic clustering technique, for which country population is one of the input variables. While Brazil and Mexico have the largest research outputs and the largest populations, this correlation between volume of research output and population size does not hold across the rest of Latin America. For example, Colombia has a small research output relative to its population size.

This concern is addressed in line 348 in the original manuscript: 'A country's socio-hydrologic cluster correlates to its representation in overall research, suggesting that, more than population size alone, a country's water and economic resources, geography, and history influence the likelihood that researchers study that country.' These findings coincide with results from Santa and Solana, 2010 who studied the overall scientific production from Latin America between 1996-2007, for which we will add a citation when this paragraph is rewritten to address concerns from RC2, detailed on page 10.

Regarding figure 10, as a Hydrologist I am proud of that result, which is totally consistent with the definition of Hydrology (Rosbjerg and Rodda, 2019), that supports the relationship of hydrology with many other disciplines.

> Authors response: As a diverse group of researchers working on water in several disciplines (e.g. geomorphology, political science, civil engineering) we are also pleased to see the interdisciplinary network shown in Figure 10. We will add this citation to the manuscript in line 425:

*While this level of connectivity is low (less than 10%), it supports the characterization of water resources as a scientific discipline where research topics are already integrated (Rosbjerg and Rodda, 2019), albeit with room for strengthened interdisciplinarity.*

An important finding is the description of the research topics presented in figures 1, 6, 7 and 8. In that sense, it is interesting to verify that statistical methods and water sampling are the

predominant methodologies part of the articles. Also, I raise the question if that groundwater could be another blind spot at the pacific side of LAC?

> Authors response: The identification of regional blind spots, such as groundwater on the Pacific coast of Latin America, is an important question which this manuscript could not address in detail given its large geographic scope. For example, the chord diagram (Figure 6) which links countries to the top 25% of studied research topics does not include 'water budget' (e.g. groundwater) or 'method' topics (e.g. water sampling). While groundwater is featured on the 'water budget' normality graph (Figure 7), the region is analyzed as a whole with all groundwater research aggregated together. The analysis displayed in this graph indicates research on groundwater is normally distributed across countries, but not normally distributed across papers, similar to other water budget topics including irrigation, lakes, and canopy interception.

To explicitly answer the reviewer's question, we returned to our data set to offer insights into the landscape of groundwater research conducted in Colombia, Ecuador, Peru, and Chile, countries in South America with only a Pacific coast.

First, the entropy of each country was calculated for subsets of topics using the Shannon-Wiener index (Shannon, 1948) to describe predictability of a random variable X with discrete probability mass function P over n outcomes. In our case, $P(x_i)$ represents the topic probabilities outputted by the topic model:

$$[H(X) = -\sum_{i=1}^{n} P(x_i) log_b P(x_i)]$$

In ecology, entropy is related to diversity through the Shannon-Wiener index. This index allows us to describe variations in ecosystems (research output) within a geographical location (country) and its overall impact on human existence (research landscape) and the environment (management of water resources). Countries on the Pacific coast of LAC have comparable diversity scores to landlocked and Atlantic-adjacent countries, with the exception of Paraguay, indicating similar distribution of research among water budget topics, which includes groundwater.

Research diversity across water budget topics

Figure SX. Diversity scores across water budget topics (e.g. groundwater, lakes) for LAC countries.

Second, we share the reviewer's interest in exploring topic- and location-specific questions and have taken several steps to allow water researchers to use our vast data set to answer their own questions. To foster transparent, reproducible and open data science, we published the non-restricted data, codes, and documentation as the R package 'wateReview'. Additionally, upon completion of our research and analysis, we applied for and received funding to construct an interactive, multilingual data visualization and communication platform using workflows developed from this research. The goal of the platform is to address the disconnect between significant science of science findings and societal need for data driven decision-making. The platform showcases how interpretable data science can significantly enhance our understanding of strengths (bright spots) and opportunities (blind spots) in water resources research and apply that knowledge for positive societal impact. We hosted three focus groups (in English, Spanish, and Portuguese) in coordination with the Global Water Partnership to include stakeholder input in the platform development. The platform, WateReview/LiterAgua/LiterÁgua, is still in development by researchers at UC Davis with expected publication in summer 2021.

Figure 4 produces a feeling of identification for a LAC water science research. It demonstrated that local problems are common to LAC community

Just for discussion and representing those who are not familiar with machine learning I wonder how much difference exists between the results obtained with the survey and the results obtained with the machine learning methodology. It would be nice to have that chance with the complementary material

> Authors response: The main points of the article are supported by findings from both our data science and social science methodological approaches, providing uniquely robust insights into the water research landscape. The following table displays the manuscript's major findings with an indication of how and where the conclusion is supported by machine learning and/or survey methodologies. This table will be added to the supplementary materials.

| Finding | Machine Learning | Survey | Line #s |
|---|---|---|---|
| Brazil dominates the water research landscape, followed by Mexico and Argentina. | x | x | 329, 352 |
| Water research on countries in the Caribbean is conducted less often than on other countries in LAC. | x | x | 353, 362 |
| Over ¾ of water research in LAC is conducted in the physical and life sciences, leading to a blind spot in the social sciences. | x | x | 356, 365 |
| Researchers collaborate primarily with colleagues in Brazil and outside of LAC, rather than with countries of similar socio-hydrologic classification. | x | x | 324, 398, 409, 439 |
| A low level of interdisciplinary research connects water researchers across the region, providing an important opportunity to build off of existing connections to expand collaboration and knowledge sharing. | x | x | 325, 425 |
| Regional knowledge sharing on research related to reservoirs and risk assessment is limited. | x | | 375 |
| Funding challenges, often related to a country's economic and political context, can inhibit research and often shape a country's research landscape. | | x | 320, 444 |

Table SX. Findings supported by data science (machine learning) and social science (survey) methodological approaches.

We also want to emphasize that there was significant human involvement to verify each step in the development of the machine learning model. Human reading was used to identify topics in >1,000 articles, as described in line 175: 'We then conducted a quality assessment of the topic models through cross-validation. For this we developed human-derived topics for the English Corpus by reading a subset of 1,428 papers from the corpus and manually identifying single-word tags based on keywords and main research topics. A similar percentage of documents were read for the Spanish and Portuguese corpora: 188 and 111, respectively.'

Additionally, the validity of the unsupervised machine learning model was assessed against human-derived topics as mentioned in line 289: 'We determined if the LDA successfully identified a relevant topic based on the top 10 occurring words which showed a 86% agreement between expert-identified topics and LDA-derived topics.'

Rosbjerg, D. and Rodda, J. 2019 IAHS: a brief history of hydrology. History of Geo- and Space Sciences. Vol 10(1) pp 109-108. https://hgss.copernicus.org/articles/10/109/2019/},

**RC2: Anonymous Referee #2, 21 Apr 2021**

The paper presents a relevant, consistent study on water resources research in Latin America and the Caribbean (LAC). According to the Authors, more than 20000 papers written in Portuguese, Spanish and English were analyzed. I would like to congratulate the Authors for the enormous effort, enabled using a powerful tool, "Machine Learning". The presented methodology is well structured and is useful for other areas of knowledge.

> Authors response: We want to thank the reviewer for the recognition of the work. While our methodology was developed to study water research in LAC, we agree that our approach and analysis tools can be easily applied to study other areas of research and hope that the publication of this manuscript alongside our publicly available and documented code base will facilitate this application.

The results and their interpretation support the presence of bright and blind research spots in LAC, indicating where and what could be developed and improved in terms of research and networks of the water community. One notes that collaboration among Brazil and other LAC countries should be augmented. Moreover, it is of concern the lack of collaboration to study international basins and the delicate results that affect all the countries involved. There are few initiatives like those within LAD-IAHR (Latin America Division of the International Association of Hydro-Environment Engineering and Research) to promote integration in LAC water community and foster Portuguese and Spanish publications, that is important for this area.

> Authors response: We strongly agree with the importance and precarity of water resources research on international basins. A review of water research and dissemination within such basins is valuable research that we will consider undertaking in the future and which we hope will be facilitated by our interactive platform, which we described in the RC1 responses. Such an effort should focus on grey literature, for which the LAD-IAHR would be an excellent resource.

Besides this general analysis, some specific comments are pointed out to rather improve the text:

In the Abstract, the first sentence could be improved. Please, state clearly the meaning of "are on particular display";

> Authors response: This sentence will be rewritten to improve clarity:

*Water resources management in Latin America and the Caribbean is particularly threatened by climatic, economic, and political pressures.*

In the Introduction section, please cite Fortunato et al. (2018) in the sequence of "Science of Science" in its first appearance. Maybe a brief explanation would also help;

> Authors response: The citation will be added along with a brief explanation, which the co-authors agree helps improve understanding:

*The state of water resources research in Latin America, including its bright spots and blind spots, can be thoroughly investigated using Science of Science: analysis of the production of science using large-scale data (Fortunato et al., 2018).*

Paragraph (lines 66-70) seems to lack of a main idea…actually, it seems unnecessary to me;

> Authors response: We acknowledge the reviewer's opinion on the last paragraph of the introduction, but would like to keep these sentences to serve as a description of the paper's structure, a practice that is common among HESS journal articles.

In the Materials section, the use of "our" methodology seems out of place. The steps for corpus collection do not seem an innovation. I would rather write: "The process of corpus collection consisted of four steps";

> Authors response: The change will be incorporated as suggested.

In lines 76-79, the Authors are explaining the first step of the method: (i) querying online databases. I suggest not to use "i, ii, iii" again to avoid ambiguity. Use a,b,c or 1,2,3 instead.

> Authors response: We understand the logic of using separate numbering systems and will change to 1,2,3 numbering for the list of criteria referenced in the comment.

In line 87, first sentence, please refer to Equations (1) and (2).

> Authors response: We will follow the HESS guidelines and refer to the equations here as Eq. (1-2).

I suggest performing a clear correspondence of the four steps stated in the first sentence of this section with the rest of it (cohesion and coherence in paragraph writing). If I´m not mistaken:

 Item "(i) querying online database" is explained in one paragraph;

Items "(ii) retrieving documents and (iii) iteratively assessing quality of the corpus and correcting bias" are illustrated in another paragraph;

The next paragraph is related to item (iii);

Item "(iv) cleaning the corpus" is explained in the last paragraph.

> Authors response: To achieve the clear correspondence requested, we will slightly restructure this section by creating a new paragraph starting at "Third", the third step in the process.

In subitem 2.2, why these databases were chosen?

> Authors response: These databases were chosen because of their international recognition, relevant indices, and global or LAC-specific applicability. We will add text to briefly mention these justifications:

*We selected the indicators from the following databases, chosen for their international recognition and global or LAC-wide breadth of data.*

In Methods section, please correct: we detailed.

> Authors response: The co-authors appreciate the suggestion but would prefer to keep the present tense when describing the paper's organizational structure, which aligns with the present tense verbs used to describe paper structure in the Introduction.

In line 167: The general metadata corresponds;

> Authors response: The co-authors prefer to keep the verb tense plural as "correspond" to align with "metadata" as a plural term. This aligns with the grammar of the previous sentence.

In the Results section, the first paragraph seems to lack of a main idea…I would rewrite it. Please make the correspondence to the respective Figure you are describing;

> Authors response: To highlight the main idea of the paragraph we will modify as follow:

*LAC countries were clustered based on socio-hydrological characteristics using hierarchical and k-mean clustering. Both clustering methods yielded similar results. The total within-sum of squares evolved after two clusters were chosen. Similarly, the average silhouette width strongly exhibited a peak for two clusters. Further inspection of clustering in principal component dimensions indicated that the cluster with Mexico and Brazil was significantly distinct from all other countries, explaining the observation of a sharp peak in average silhouette width. However, validation metrics exhibited optimal null values of APN and ADM for two or three clusters. In addition, AD and FOM were lower for three clusters than for two. Based on these results, we chose three clusters to describe the grouping of countries based on their socio-hydrologic variables.*

In lines 285 and 289, please explain the abbreviations in their first appearance in the text;

> Authors response: These results describe the outcome from the validation metrics used to assess the stability of the clustering, which are explained when they first appear in the manuscript in lines 157-162, however we will change the acronyms to the complete names to improve clarity.

In lines 314 and 315, please rewrite this first sentence;

> Authors response: The sentence will be rewritten as follows to redefine the term "location":

*For predicting location of the country of study of each paper, the random forest analysis outperformed every other model with a mean multiclass AUC of 0.99 and a mean accuracy of 96% (Fig. 3).*

In line 349, please break the sentence into several sentences to improve its understanding;

> Authors response: We will rewrite the sentence into several sentences to clarify the message:

*A country's socio-hydrologic cluster correlates to its representation in overall research, with Cluster 1 (Brazil and Mexico) receiving the most research, followed by Cluster 2, then Cluster 3, which includes most of Central America and the Caribbean. Although population size likely affects each country's representation in the overall research output, it does not precisely correlate with research volume. We therefore expect that other factors used to define the country clusters (e.g. a country's water and economic resources, geography, and history) influence the likelihood that researchers study that country.*

In lines 375-380, I was confused with the text. Could you rewrite it, please?

> Authors response: We will rewrite this text to improve clarity:

*The least normality is seen in two topics of great importance for water management: reservoirs and risk assessment. Both topics have normality values far below 1 across both countries and documents, suggesting poor representation of these topics on a broad scale (Fig. 8). These are alarming blind spots given the importance of reservoirs for water supply reliability and their impacts on local communities and ecosystems. Furthermore, a lack of research in reservoirs and risk assessment has troubling future consequences as climate change increasingly acts as a risk multiplier, including for reservoir operation. Although it is possible that information on reservoirs and risk assessments exists in grey literature, such as government reports, university publications, and conference proceedings, this information is not as accessible as scientific publications and limits regional knowledge-sharing. These findings confirm similar results in a country-specific analysis of water research opportunities in Brazil (Paiva et al, 2020).*

In subsection 5.5, due to the existence of a lot of information, I was also confused with the analysis and correspondence (in particular) to Figure 10b. Could the Authors further clarify this correspondence in the Figure?

> Authors response: To improve correspondence of the analysis to each subfigure, an extra sentence will be added for clarity when introducing the Figure 10:

*We assessed the connectivity of water research throughout the region by aggregating research from all countries by topic (Fig. 10). Separate networks are presented to illustrate connectivity between the five general topics (Fig. 10a), and specific topics (Fig. 10b).*

**CC1: Pedro Luiz Borges Chaffe, 06 Apr 2021**

This paper was a very pleasant read. The literature review is quite impressive and reveals important issues on water related research in LAC countries. We see it as an important step for building a common research agenda for LAC and fostering collaboration as highlighted in the paper results.

Our brief commentary is motivated by some of the similarities we found in a synthesis exercise with the Brazilian water resources community (i.e., Paiva et al., 2020). Paiva et al. (2020) is mainly based on grey literature – i.e., the proceedings of the XXIV Brazilian Water Resources Symposium 2019 – which complements a limitation pointed out in this paper.

> Authors response: We thank the commenter for bringing our attention to Paiva et al. (2020). This article is an excellent complement to our manuscript and we will reference it in several places, including line 37 which will be rewritten to:

*Recent review papers are limited to a geographic area (Owusu et al., 2016), individual components of the water budget, such as a watershed (Dobriyal et al., 2012), particular methodology (Plummer et al., 2012), specific water user (Ran et al., 2016), or small sample of documents (Endo et al., 2017).] Paiva et al. (2020), for instance, conducted a review of 250 conference papers from the 2019 Brazilian Water Resources Symposium to better understand major advances and challenges in Brazil's water science.*

DeVincentis et al. (2021) is right that many of the innovations in the water resources community may appear only in grey literature. We believe that relevant experience-based innovations – coming mostly from practitioners outside of academia and dealing with pressing issues (such as in the case of reservoirs and risk assessment) – are usually reported in symposium proceedings, and may not be rigorously documented in peer reviewed scientific papers. Moreover, Paiva et al. (2020) points out that most of research in Brazil focused on water resources practice, with more emphasis on methods and estimation, quantification, and prediction of water-related phenomena, while there is a clear opportunity for more research on processes and phenomena comprehension. It would be very interesting to understand if the same pattern is seen throughout LAC.

> Authors response: Our results describing LAC as an entity are inherently limited given the diversity throughout the region. The importance of describing country-specific water research opportunities, such as those described by Paiva et al (2020), is paramount to conducting and funding regionally-relevant water research. Country-specific analysis, while out of scope of this manuscript, can be performed using our data set and code base and we hope such effort will be undertaken.

In a way, many of the same problems observed in the LAC context are also expressed at the Brazilian national scale. Regarding the issues of collaboration, or the lack thereof, it is a common theme even inside the same country. In Paiva et al. (2020), we found that there is an intense network of collaboration mainly between academic institutions. However, there is regional fragmentation, possibly due to geographical convenience, legacy, or common interest on regional scale water resources issues. Brazilians that study abroad tend to maintain these collaborations, so a lot of the co-authorship with American and European researchers found in this paper might be due to legacy from graduate level training. Moreover, Brazil is a destination for several graduate students from South American countries, which might explain its high frequency in collaborations within LAC even being the only Portuguese speaking country. In order to foster collaboration among LAC countries, we think there could be a common South and/or Central American association and events which could congregate researchers working on water issues in the region. In Brazil, for example, most water researchers are associated with the Brazilian Water Resources Association (ABRhidro), and gather bi-annually for the Association Symposium.

> Authors response: This is an interesting comment that provides valuable insights into potential causality of Brazil's dominance and centrality in the research landscape. We will add this context and another reference to Paiva et al. to our discussion on Brazil's role in water research in the paragraph beginning on line 398.

*Despite being the only Portuguese-speaking country, Brazil was the country most often listed in collaborations within LAC (17%). Brazil's prominence may be partially explained by the legacy of relationships formed during graduate level training when Brazilian researchers study abroad and when graduate students from other South American countries study in Brazil. Despite its greater connectivity, the review of 250 water science papers presented at the 2019 Brazilian Water Resources Symposium still found a lack of a common scientific agenda within the country, and a need for more interdisciplinary research and collaboration with international communities, "especially with other Latin American countries with shared water issues" (Paiva et al. 2020).*
*A quarter of research collaborations involve non-LAC researchers...*

As the paper eloquently points out, an opportunity exists to form a strong community of researchers and strengthen research impact through knowledge sharing. We also believe that we should enhance two-way sharing of knowledge and efforts on water sciences development, especially with other LAC countries with shared water issues. We should combine our experiences to actively contribute to the tackling of global water issues.

We congratulate the authors for this interesting contribution and hope that our comments can be useful for enhancing the discussions of this paper.

> Authors response: We greatly thank the writers for this thoughtful commentary and useful recommendations.

Pedro Chaffe (pedro.chaffe@ufsc.br), Federal University of Santa Catarina, Brazil

Rodrigo Paiva (rodrigo.paiva@ufrgs.br) , Federal University of Rio Grande do Sul, Brazil

**CC2: Daniel Prieto Garra, 03 May 2021**

REVIEW OF THE BRIGHT AND BLIND SPOTS OF WATER RESEARCH IN LATIN AMERICA AND THE CARIBBEAN

By Daniel Prieto Garra1

1 Dr (Wageningen, The Netherlands), MSc (Wageningen, The Netherlands) Agricultural Engineering (UdelaR,Uruguay). Former National Coordinator of the Water Resources Research Program (2001-2018) of the National Institute for Agricultural Technology (INTA) from Argentina. Freelance consultant (2019-2021).

Very valuable and excellent work!!!. It added both very valuable innovative methodology for meta-analysis works and a great information from LAC research works on water resources.   Of course, I have nothing to add to your excellent innovative methodology. I am just adding some comments from my point of views that can help the discussion of your findings and that I hope could be useful for the participatory review of your article (a review process that I celebrate and congratulate you).

**Introduction**

I agree with most of your statements in this section since they are close of an objective characterization of the water resources and its problems in LAC

**Line 20** and the following. Just as simple information, I would like to highlight than Argentina, like Brazil and Chile, is another clear example of the uneven distribution of the Water Resources in the continent. Water rich areas, are only 24% of the country territory, own 82% of the surface water resources ...Arid and semi-arid regions, the complement 76 % of the country area has only has 18% of the surface water resource and 80% of them are in the Patagonia. In Argentinean case, 72% of its population live in rich water areas where the population density reach 21 inhabitants/km2, while in the semi-arid and arid region it is only 5.9 inhabitants/km2.

The above does not mean that Argentina has water resources problems only in the semi and arid region. Rich water areas suffer periodic floods that affect mainly more poor people that live in vulnerable areas of the big cities and also suffers of frequent dry short period during the summer that affect mainly the agricultural production the principal base of Argentine economy. Water availability and water quality are of course the main water related problem in the Arid and Semi-arid regions. Natural derived water resources problems are amplified in both area by a lack of a professional water management due financial constraints and institutional and human resources weakness.

> Authors response: We truly appreciate the nice words and recognition of the work and thank the reviewer for taking the time to provide this review. We will alter the sentence in line 25 to provide an example of high population density:

*LAC is among the most urbanized regions in the world where population densities in water-rich regions can be many times higher than arid regions, such as in Argentina. These high-density areas face particular vulnerability to water quality and supply reliability (Kim and Grafakos, 2019).*

**Line 33**. "Water management is a relatively young field of study and suffers from a lack or interdisciplinary and integrative perspective........." Crucial statement from my point of view.

**Line 59**. How do you explain the low number of response to your invitation to corresponding authors?

> Authors response: We attribute the relatively low percentage (10%) to the uncertainty of internet polling, the quantity of emails that were unsuccessful (failed address or email blocked as unknown sender), and because online surveys often achieve lower response rates than traditional paper surveys (Nulty 2008). Additionally, although the survey was available in multiple languages, a language barrier may be partially responsible for the response rate. Still, we accepted the ~10% survey response rate as sufficient considering that 1,969 is a fantastic number of responses.

 **Subsection 4.2.2**

**Line 320 to 325**. In my opinion this lines pointed out one of the main problems of the LAC′s researchers. The unequal distribution of funds within the country (always large research group of the rich areas of the country access to fund, even those competitive ones) and the lack of continuity of the political and economic context that affect research support policies in our countries.  Due these points, LAC researcher strategies focus on look for collaboration with researchers from USA and especially from Europe, (more funds can be find in this area). I would add that there is a lack of incentive from our governments to support and strength LAC′s research network. I limited positive example that I know, was the strategy for foment regional research network of the PROCISUR (a cooperative program of the agricultural research institute from Argentina, Brazil, Chile, Paraguay and Uruguay) during the first decade of 2000′s. However as soon the institutes had financial problem related with neo-liberal policies implement by the government of the region it was discontinued.

> Authors response: We agree that LAC experiences a serious issue of not only unequal distribution of funding between countries, but also an inequity of funding opportunities within countries between wealthier and less-wealthy regions. The inevitable internal struggles and

research in-country disparities caused by this inequity are a hugely important issue which our data set and methodology could not shed light on, but which we hope future work will address.

**Lines 357 to 359**. I agree with your finding about water research in social sciences based on the meta review of peer-reviewed literature. Nevertheless I consider that your focus on peer-reviewed literature bias your findings. From my personal experience, I have the qualitative idea that research on socio-technical problems on water management in LA has increased sharply during the last decade. Findings of this research (mainly study cases and/or participative research with local actors) are seldom submitted to peer-reviewed journals. They are mainly reported in books (see Boelens et al, 1998, 2002, 2005), regional congress, or directly translated to diffusion publications.

> Authors response:  This point will be added to the discussion around lack of social science research in line 357 when describing results from the chord diagram (Figure 6):

*The lack of water research in social sciences may reflect a combination of low publication rates (compared to physical sciences), disciplinary preference for publishing in books rather than peer-reviewed articles, a historical framing of water management as a purely technical discipline (Callaghan et al.,2020), and financial resources from governments biased towards physical sciences and engineering.*

From the above I would recommended you to explore this type of publication in a future update of your brilliant, useful and great works.

> Authors response: We again thank the reviewer for the nice words and will explore other types of publications, such as conference proceedings, in the future as we agree and know from experience that in LAC there are many reports with vast information that are not translated into peer-reviewed publications.

**Additional considerations**

During this review process, the authors discovered a typo in section 5.4. The title will be corrected to 'Brasil é nùmero um'.

Lastly, it was brought to our attention that the term 'blind spot', featured predominantly in the manuscript to describe under-studied research topics, could be perceived as insensitive to individuals with vision impairment. A discussion amongst the co-authors identified valuable reasons to both remove and keep the reference, citing online forums where ableist language is discussed.

The consensus expressed by visually impaired individuals was that ableist language consists of terminology which is derogatory towards that group, but that the term blind spot does not inherently fall into this category. Additionally, multiple comments mentioned the harm of labeling language as ableist by people who are not members of the affected community because it could create undue confusion around the true harm of ableist language. Furthermore, the term blind spot which refers to the unintentional inability to see a few things, which affects all people and is a universal expression in the English language, most accurately conveys our understanding of the research gaps highlighted in this study.  For these reasons, we believe that this is the most appropriate title, however we bring it to the editor's attention for full disclosure and are willing to discuss alternative titles if the editor feels differently.